# Mass-Produced Cu Nanoparticles as Lubricant Additives to Enhance the Tribological Properties of DLC Coatings

**Nan Li** [1,*], **Mingchang Wang** [1] and **Zhiguo Wu** [2]

1   Suzhou Nuclear Power Research Institute Co., Ltd., Shenzhen 518008, China
2   Institute of Nanomaterials Application Technology, Gansu Academy of Science, Lanzhou 730000, China
*   Correspondence: linan6052960@126.com; Tel.: +86-18138803107

**Abstract:** In this paper, Nano copper (Cu) particles with a core-shell structure and good spherical shape were prepared by an innovative method called mass-produced nanoparticles (MPNP). The prepared Cu nanoparticles have good dispersibility and are agglomeration-free in Pao oil. In particular, the effects of nano-Cu particles with different mass fractions on the tribological properties of the steel against diamond-like carbon (DLC) coating were studied systematically. The results showed that the nano-Cu particles with the mass fraction of 0.1 wt.% and the steel/DLC friction pairs had good synergistic lubrication. The friction mechanism of the metal nano-particles and carbon-based coatings mainly depends on the physical effects such as nano-bearing and nano-filling of the nano-particles, which has little correlation with the shear film formation of the metal nano-particles. Therein, the surface polish behaviors of the metal nano-particles and carbon-based coatings are the key to bringing the nano-bearing mechanism of nano-particles into full play. Therefore, the Cu nanoparticles prepared by MPNP show excellent tribological performance and possess broad prospects in the fields of lubricant additives.

**Keywords:** Nano-Cu particles; diamond-like carbon coating; solid-liquid lubrication; friction and wear





## 1. Introduction

Solid-liquid lubrication technology is an important method to reduce the material friction coefficient and wear rate under extreme working conditions [1,2], which improves the lubricating effect by using the synergistic effect between the lubricating materials of solid and liquid. The carbon-based coatings have the characteristics of high hardness, low friction coefficient, excellent wear resistance, and self-lubrication, and it exhibited the great application potential in reducing friction and wear of mechanical parts [3–7]. However, owing to the surface of diamond-like carbon (DLC) coatings have special physical and chemical properties, they are greatly different in friction and wear properties from the traditional metal tribological pairs when compounded with the liquid lubricants. The oil lubrication behaviors of DLC coatings and solid-liquid synergistic lubrication mechanism have been widely concerned. Recently, the influence of lubricant additives on the solid-liquid composite lubrication behavior of DLC coating is one of the main research directions [8,9].

Although a certain degree of tribo-chemical reaction exists between DLC coating and additives, the degree of reaction is low and it is limited by the friction effect of the additives [10–12]. With the increasingly stringent environmental protection requirements, traditional additives including sulfur, phosphorus and so on will be eliminated gradually [13]. Although environmentally friendly lubricants such as Glyceryl monooleate (GMO) and DLC coating have a good synergistic effect the problem of oxidative degradation exists due to its molecular structure containing active groups such as double bonds, ester groups, and hydroxyl groups, especially the water adsorption layers are generated by decomposition dominate its friction reduction mechanism, thus it is not a long term effective and reliable lubrication method.

The interaction and lubrication mechanism between lubricating oil additives and metal friction pairs were studied in nano-additives [14–16]. Kalin et al. [17] found that adding a small amount of $MoS_2$ nanotubes (2 wt.%) to the lubricating oil can significantly reduce the friction coefficient of the DLC coatings under the boundary lubrication by about 40% through a series of studies on $MoS_2$ nanotubes. Zeng et al. [18] reported the phenomenon of the super lubrication ($\mu < 0.01$) in $Si_3N_4$/DLC friction pairs by using boron nitride nanoparticles as additives, and pointed out that the super lubrication is caused by the micro bearing effect of boron nitride nanoparticles and slide of the lamellar structure in the particles. At present, most of the investigated nano-additives are focused on the traditional solid lubricating materials with one-dimensional and two-dimensional nanostructures such as $MoS_2$, carbon-based, hexagonal boron nitride (h-BN) and so on [2,19], and there are few studies on metal nano-particles as additives of the carbon-based coatings solid-liquid composite lubrication system.

The interaction and the lubrication mechanism between the metal nano-particles and carbon-based coatings have few reported have rarely been reported. The metal nanoparticles are more active and more likely to form tight bonds with carbon coatings. As a kind of soft metal with face-centered cubic structure, Cu nanoparticles have been favored by researchers because of their advantages of low shear force, excellent ductility, low melting point, low phase transition temperature and more environmental protection, and reducing friction and wear in lubricating oil additives. Cu nanoparticles can be sheared to form the lubricating films on metal surfaces, while the formation of structured low-shear friction films on the surface of DLC coatings by Cu nanomaterials is unknown so which needs to be further investigated.

In this paper, nano-Cu particles were produced by a new technology called mass-produced nanoparticles (MPNP), which has many distinctive advantages, including the high-production rate of more than 300 g/h, the high-utilization rate of raw materials and high yield. In addition, nano-Cu particles have very good uniform particle size, outstanding dispersion and excellent sphericity. The effect of various contents soft metal nano-Cu additives on the tribological properties of the steel/DLC friction pairs was analyzed systematically. Exploring the interaction mechanism between metallic Cu nanoparticles and DLC coatings, and initially putting forward the proposal for the construction of nano-additive suitable for steel/DLC friction pairs. Based on the above discussion, the effect of nano-Cu particles in improving the tribological performance of DLC coatings and their synergistic lubrication mechanism was also investigated in detail.

## 2. Materials and Methods

### 2.1. Preparation and Characterization of Nano-Cu Particles

The metal nano-Cu particles an innovative technology called mass-produced nanoparticles (MPNP) were prepared in the study. Oxygen is slowly added into the vacuum chamber, and then a dense oxide layer of several nanometers in thickness is formed on the particles' surface, which contributes to the preparation of high performances metal nano-particles. Owing to the ion beam can deliver high energy continuously, high surface powder density and five ion beams touching the raw material at different locations. Copper powders (99.9% pure) in the procedures were used as raw materials to prepare copper nanoparticles. A large of the raw materials will be evaporated rapidly and produce a large of the metal nano-particles with high activity and high surface finish without wasting, which can realize industrialization, and the productivity outnumbers 300 g/h [20]. The collection device should be kept below 25 °C. Because of a large temperature gradient between the steam generation area and the manifold, the metal vapor cools rapidly through the cooling zone. Therefore, the monodisperse nano-powders were prepared via nucleate rapidly and uniformly and quench. The preparation device has advantages such as the simple structure, easy operation and no limits on the shape of metal raw materials. The steps of the obtained core-shell structures are as follows. Step 1: In the beginning, the high-purity material containing copper was placed in the graphite crucible as raw mate-

rials. Step 2: Then the chamber of the experimental equipment was in a vacuum state of $1 \times 10^{-3}$ Pa. Step 3: Through the specific energy transformation and temperature field, raw materials transformed into a flowing (liquid, gas, or ionic) state and rapidly cooled through the cooling zone then nucleate and agglomerate to grow into Cu nanoparticles. Step 4: After a certain period of time, the chamber was filled with gas to atmospheric pressure, and a few minutes later the Cu nanopowders were collected. More preparation details can be found in our previous work [20,21].

The size, morphology and dispersion, and Energy Dispersive Spectroscopy (EDS) analysis of nano-Cu particles were observed by scanning electron microscopy (SEM, MIRA3, TESCAN, Brno, Czech) with the specific working conditions of high vacuum mode ($<9 \times 10^{-3}$ Pa), accelerating voltage of 15 kV, probe current of 40 nA, and the maximum resolution of 1.2 nm. The crystal structures of metal nano-particles were characterized by X-ray diffractometer (XRD, D/Max-2400, Rigaku, Tokyo, Janpan) at $2\theta = 10–90°$ of measurement angle range. Then the samples were degassed at 100 °C for 12 h, and the nitrogen adsorption and desorption isotherm were obtained at 77.35 K. Using the Brunauer-Emmet-Teller (BET) method, the specific surface area (Sw) of the nano-Cu particles was obtained, which also can be used to calculate the crystalline grain size (Ds).

### 2.2. Deposition of DLC Coatings

An unbalanced magnetron sputtering equipment (UDP650, Teer, Hartlebury, UK) was used to deposition of DLC coatings on the surface of the 304L stainless steel. First of all, the 304L stainless steel sheets were ultrasonically cleaned with acetone and alcohol for 15 min and dried with dry nitrogen, then put them on the sample holders of the vacuum chamber. Turned on the air extraction system of the equipment and the vacuum was pumped to $1.0 \times 10^{-6}$ torr, then injected argon and adjusted the pressure to $1.6 \times 10^{-3}$ torr. Argon ions are excited to clean the substrate surface for 35 min when the bias increase to $-500$ V. As shown in Figure 1, the bias voltage was adjusted to $-300$ V, and the DC power was switched on to adjust the current to 3.0 A. Two Cr targets (purity of 99.99%) were sputtered to prepare a Cr transition layer to improve the adhesive between the carbon-based coating and the substrate. DLC coatings were deposited on the chromium (Cr) interlayers by sputtering two graphite targets (purity of 99.99%). After deposition, the unbalanced magnetron sputtering equipment was turned off and removed the DLC coatings from the vacuum chamber after cooling for 2 h. The specific mechanical properties are shown in the previously published literature [22].

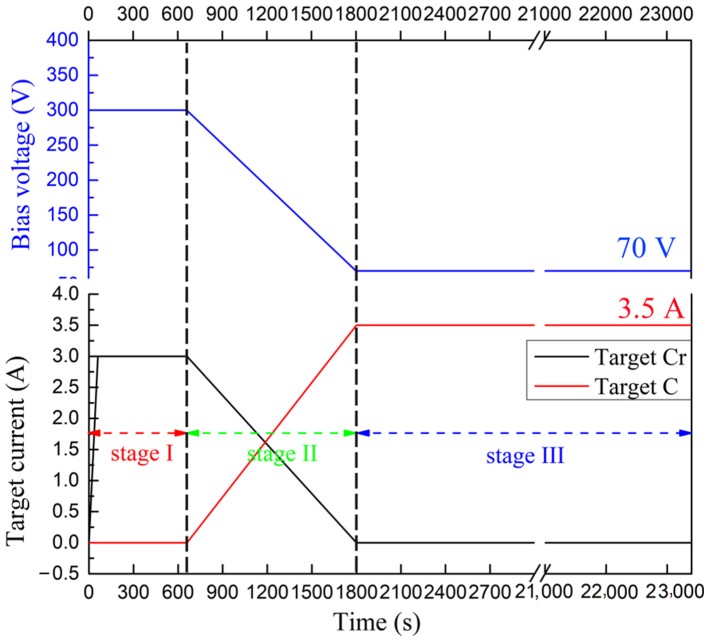

**Figure 1.** Schematic diagram of DLC coatings deposition process.

### 2.3. Tribological Properties

The tribological behaviors of deposited DLC coatings in polymerized alpha olefin (PAO) oil with different mass concentrations of Nano Cu particles were evaluated by SRV-IV tribo-tester (Optima, Schwäbisch Hall, Germany) in pin-on-disk reciprocating mode. The schematic diagram of the tribo-tester is shown in Figure 2. The test environment is open air with room temperature of about 20~25 °C, the applied load is 100 N, the test time is 30 min, the relative humidity is 25~30%, and the mating ball is a GCr15 steel ball with a diameter of 10 mm. After tests, the film wear tracks were measured by a non-contact two-dimensional surface profiler (micro XAM, KLA-Tencor, Milpitas, CA, USA), and then wear rates were obtained according to the formula:

$$\text{Wear rate} = \text{Wear Volume}/(\text{Applied load} \times \text{total sliding distance}).$$

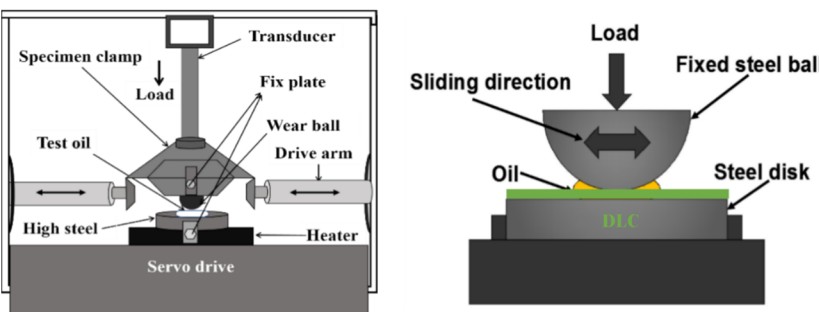

**Figure 2.** Schematic diagram of SRV-IV tribo-tester and steel/DLC friction pair.

After the friction test, the DLC coatings were cleaned with acetone to remove the residual oil, and then the morphology and chemical composition of film wear tacks and ball wear scars were analyzed by scanning electron microscopy (SEM, MIRA3, TESCAN, Brno, Czech) with the specific working conditions of high vacuum mode ($<9 \times 10^{-3}$ Pa), accelerating voltage of 15 kV, probe current of 40 nA, and the maximum resolution of 1.2 nm. The morphology and roughness changes of DLC films before and after friction tests were observed by atomic force microscope (AFM 5500M, Benyuan, Guangzhou, China). The tip of AFM was a $Si_3N_4$ tip with a radius of 10 nm. The composition and structure changes of DLC films before and after friction tests were analyzed by Energy dispersive spectroscopy (EDS) and Raman spectroscopy.

### 3. Results and Discussion

#### 3.1. Characterization of Metal Cu Nanoparticles and DLC Coatings

The SEM image and EDS result of Cu nanoparticles are shown in Figure 3a,b. Most of the metal Cu particles are spherical, monodisperse, and non-agglomerated, as illustrated in Figure 3a. EDS result shows that, as shown in Figure 3b, the Cu nanoparticles are mainly composed of 96.28 at.% Cu and 3.72 at.% O. Figure 3c displays particle size distribution statistics of Cu nanoparticles. The particle size of Cu nanoparticles is mainly in the range of 50–150 nm with an average particle size (Dn) of 86.11 nm, and 95.83% of metallic Cu particles are smaller than 200 nm. According to the XRD pattern, as shown in Figure 4, the relative intensities of the diffraction peaks corresponding to the diffraction angles of the Cu nanoparticles are consistent with the XRD data of the strongest peaks shown on the powder standard card, respectively [23]. In addition, the XRD pattern also shows a very weak of $CuO_2$, which is consistent with the results of EDS.

Figure 5 presents the cross-sectional morphology and surface morphology of the deposited DLC coatings. As shown in Figure 5, the DLC layer thickness of approximately 4.6 μm is observed. The DLC coating surface shows a dense structure consisting of a number of hemispheres, which corresponds to the columnar structure of the cross-sectional morphology. Based on our previous work [22], the DLC coatings prepared have excellent

mechanical properties, the hardness and elastic modulus are 12.2 GPa and 158.5 GPa, respectively, with the critical load (Lc1) is 48 N.

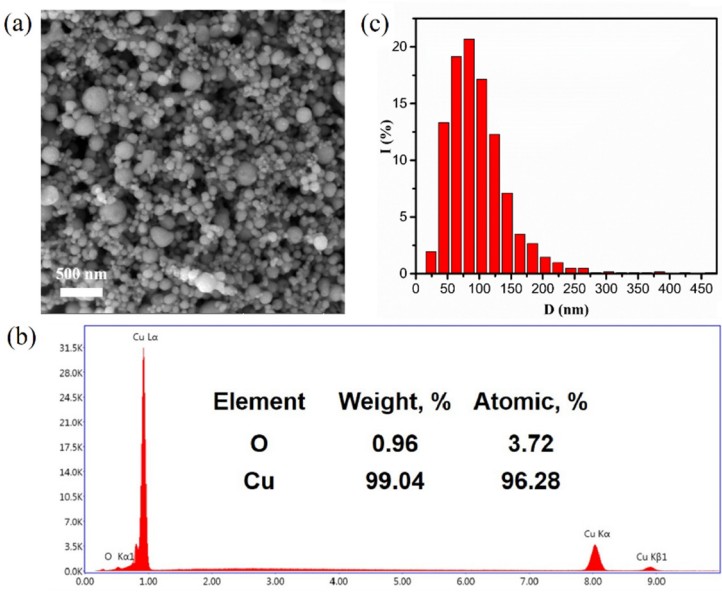

**Figure 3.** SEM image (**a**), EDS results (**b**) and Statistical diagram (**c**) of particle size of Cu nanoparticles.

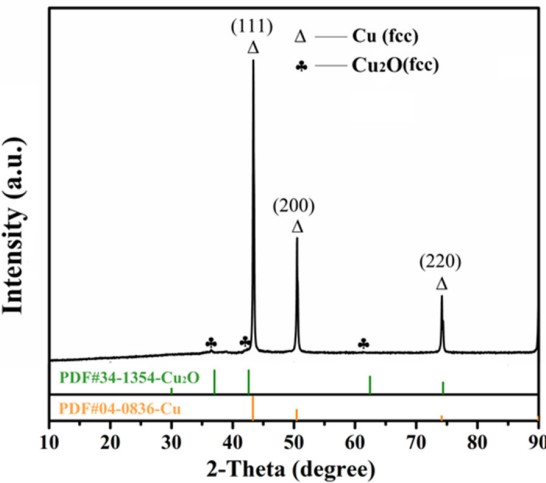

**Figure 4.** XRD diffraction of Cu nanoparticles.

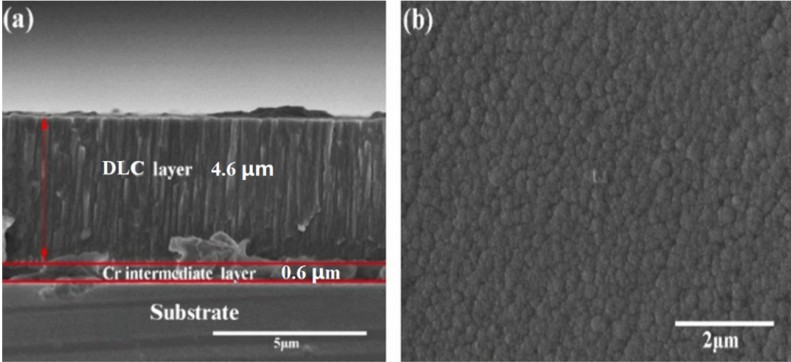

**Figure 5.** Cross-sectional morphology (**a**) and surface morphology (**b**) of DLC coating.

### 3.2. Effect of Cu Nanoparticles on the Friction and Wear Properties of Steel/DLC Tribo-Pairs

In order to better discuss the effect of metal nanoparticles on the tribological properties of lubricants, the tribological properties of prepared metal nanoparticles as base oil additives are assessed and the results are represented in Figure 6. The friction coefficient and wear rate of the steel/DLC friction pairs vary with the amount of Cu nanoparticles added, but do not change significantly with the concentration of Cu nanoparticles, remaining at around 0.13. Besides, the wear rate of DLC coatings is significantly lower compared to pure base oil PAO6 with a wear rate of $3.12 \times 10^{-8}$ mm$^3$/(Nm). The wear rate of DLC coatings initially decreases gradually and then increases as the concentration of Cu nanoparticles increases. At low concentrations (0.05 wt.% Cu), there is no friction film formed between the Cu nanoparticles and the steel/DLC friction pairs surface, only the steel balls are in contact with the substrate. At high concentrations (1.5 wt.% Cu), the enriched Cu particles hinder friction and increase wear. The lowest friction coefficient and wear rate are measured when the concentration of Cu nanoparticles is 0.1 wt.%.

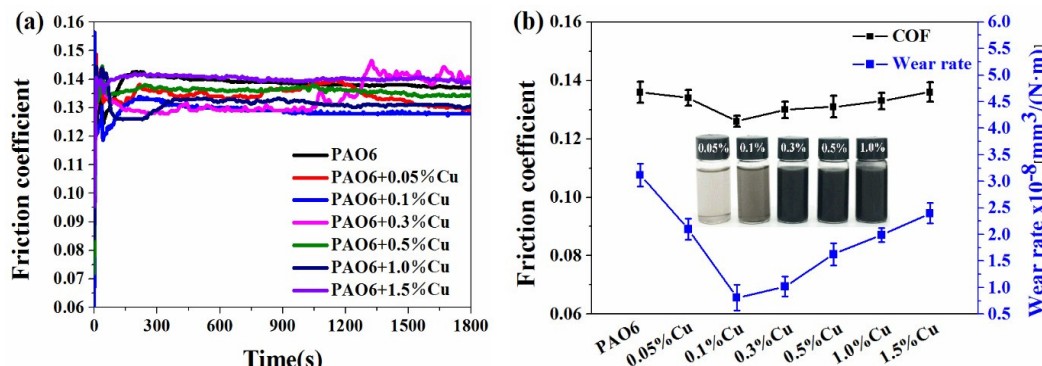

**Figure 6.** Friction curves (**a**) and wear rates (**b**) of steel/DLC friction pairs with different addition of Cu nanoparticles.

### 3.3. Discussion

Figure 7 shows the SEM images of the wear tracks of Cu nanoparticles with different additions on the surface of the steel/DLC friction pairs. Wear track under pure PAO6 lubrication is presented in Figure 7a. There are numerous furrows and grooves on the wear track, indicating that the abrasive wear dominated the wear mechanism. Figure 7b–g shows the morphological characteristics of the wear track under different concentrations of Cu nano-additive lubrication. The inside of the wear surfaces becomes smoother and the depth of the wear track becomes shallower. Especially when Cu is added at 0.1 wt%, the values of friction coefficient and wear rate of the steel/DLC friction pairs are minimum and the lubricity is best. From Figure 7b, it is seen that the surface morphology of the DLC coating will significantly influence the distribution state and dynamic behavior of the nanoparticles in the contact area, the initial surface micro-bumps of DLC coatings are not conducive to rolling of nanoparticles between contact interfaces. However, the micro-bumps on the surface of the DLC coating were removed by the polishing effect of the Cu nanoparticles during the friction process. The surface morphology of the coating is transformed into a highly consistent distribution and smooth platform morphology, which facilitates the nano-bearing effect of nanoparticles that improves the tribological properties.

Furthermore, the SEM images of the wear scars on the counterpart steel balls with different additions of Cu nanoparticles in PAO6 base oil were characterized. Figure 8a shows severe wear on the counterpart surface under pure PAO6 lubrication, with many deep grooves. Figure 8b–g shows shallow grooves and a few wear debris on the wear surface lubrication with different content of Cu nanoparticles in PAO6. The wear scar of the counterpart presents the smallest diameter and wear is minimal when the Cu additive is 0.1 wt.%. Based on the above results, the Cu content of nano-additive with 0.1 wt.% obtain the best wear resistance and lubrication performance compared to PAO6 oil, thus showing

the best tribological properties in steel/DLC friction pairs. The addition of Cu nanoparticles is beneficial to reducing the abrasive wear of friction pairs, filling micro-pits, and grooves on wear surfaces to improve wear resistance. There is no chemical reaction between the Cu nanoparticles and the DLC coating during the friction process. The improvement of the tribological properties of solid-liquid composite systems of DLC coatings by Cu nanoparticles is mainly based on the physical mechanism of the nanoparticles, which reduces the friction and wear through the function of "micro-bearings", filling micro-pits and boundary lubricating films.

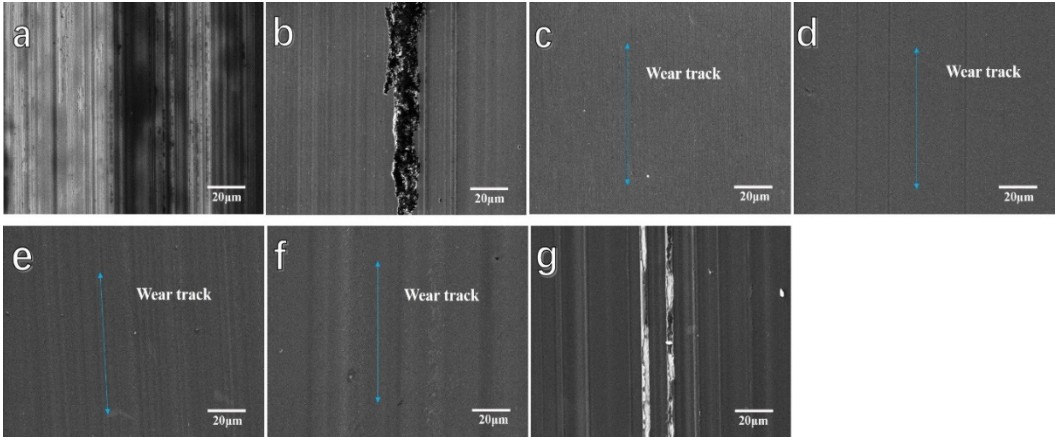

**Figure 7.** SEM images of wear tracks on the surface of DLC coatings lubrication with different content of Cu nanoparticles in PAO6: (**a**) 0 wt.%, (**b**) 0.05 wt.%, (**c**) 0.1 wt.%, (**d**) 0.3 wt.%, (**e**) 0.5 wt.%, (**f**) 1.0 wt.%, (**g**) 1.5 wt.%.

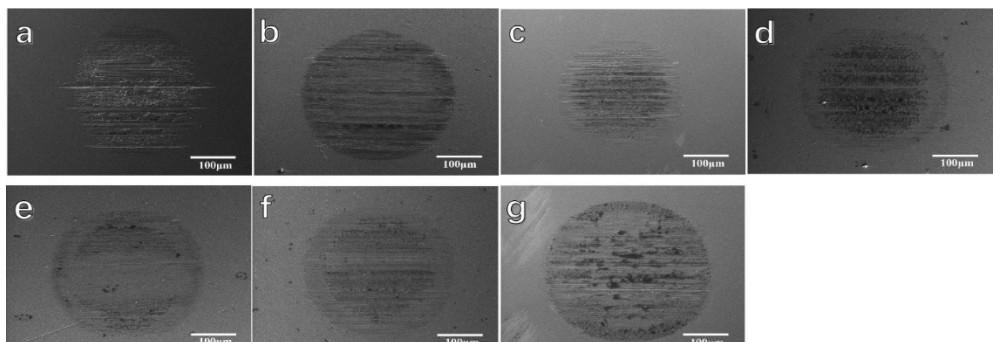

**Figure 8.** SEM images of wear scars on the surface of mating balls lubrication with different content of Cu nanoparticles in PAO6: (**a**) 0 wt.%, (**b**) 0.05 wt.%, (**c**) 0.1 wt.%, (**d**) 0.3 wt.%, (**e**) 0.5 wt.%, (**f**) 1.0 wt.%, (**g**) 1.5 wt.%.

The contact resistance (ECR) of the friction pairs reflects the state of the interface and is sensitive to the composition and microstructure of the friction pairs. Figure 9 shows the ECR curves with time for the steel/DLC friction pairs under PAO6 and PAO6 + 0.1 wt.% Cu lubrication. It is concluded that the ECR was high for pure DLC films, at around 2.6 KΩ. The electrical conduction mechanism in DLC coatings is the transport of electrons in $sp^2$ carbon bonds through a tunnel penetration process or by thermal excitation, and DLC coatings with high $sp^2$ content generally have a high electrical conductivity. According to Figure 9, the value of ECR is significantly reduced from 2.6 KΩ to 1.8 KΩ when Cu nanoparticles stay on the interlayers surfaces of DLC coatings. It implies that the higher ECR value of DLC/steel friction pairs mainly depends on the influence of the actual contact area.

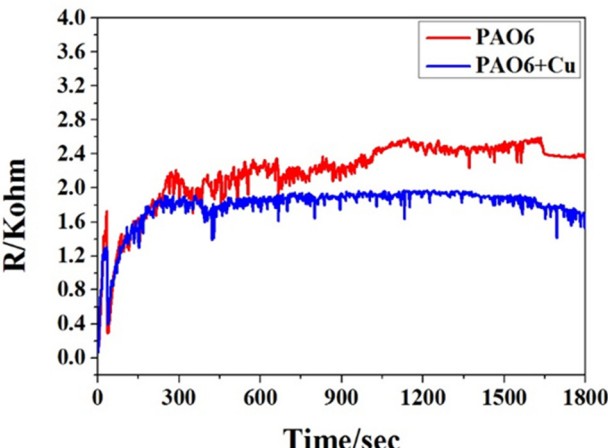

**Figure 9.** Variation curve of surface contact resistance of steel/DLC friction pair lubrication with PAO6 and PAO6 + 0.1 wt.% Cu.

Figure 10 shows the Atomic Force Microscope (AFM) images of wear scars for steel/DLC friction pairs. Figure 10a presents the deep grooves and high roughness on the wear surface due to abrasive wear under pure PAO6 oil lubrication. Figure 10b–g displays a significant reduction in the roughness of wear surfaces, and the number of furrows decreased and a smoother surface with the addition of different concentrations of Cu nano-additives compared with pure PAO lubrication. In particular, the wear surface is highly homogeneous with micro-platform morphology at 0.1 wt.% Cu concentration, as shown in Figure 10c.

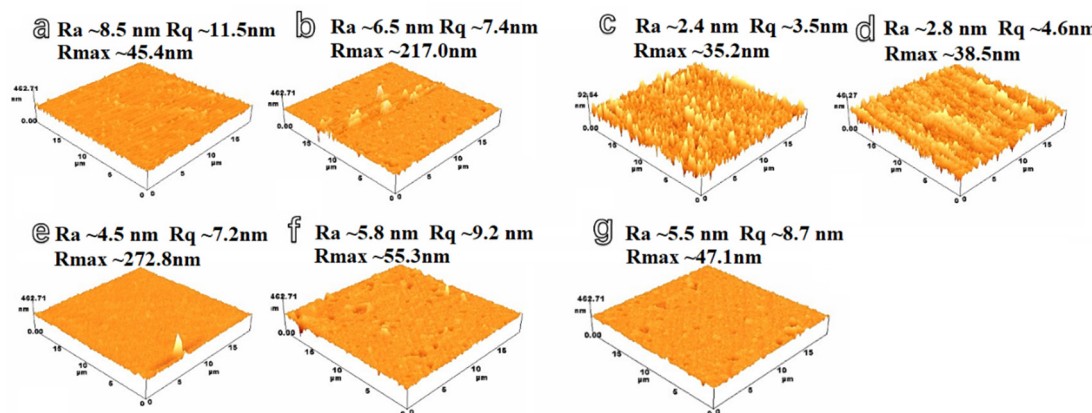

**Figure 10.** AFM images of wear tracks on the surface of DLC coatings lubrication with different content of Cu nanoparticles in PAO6: (**a**) 0 wt.%, (**b**) 0.05 wt.%, (**c**) 0.1 wt.%, (**d**) 0.3 wt.%, (**e**) 0.5 wt.%, (**f**) 1.0 wt.%, (**g**) 1.5 wt.%.

The Raman spectra of the DLC coatings before and after tribological test are given in Figure 11. All spectra show a typical peak shape for amorphous carbon materials, an asymmetrical broad peak between 1100 and 1750 $cm^{-1}$. This indicated that the DLC coating retains amorphous structure during the friction process. In both pure PAO and PAO with Cu nanoparticles, the G-peak of the DLC coating shifts towards the high wave number after friction, and the $I_D/I_G$ value increases. In the Raman spectrum of DLC coatings, the position of the G peak is related to the telescopic vibrations of the C=C bond in the amorphous carbon skeleton structure, the stronger the C=C double bond vibration, the higher the wave number of the G peak. The intensity ratio $I_D/I_G$ is related to the size of the graphite clusters in the system [24–26], the larger the cluster of $sp^2$ carbon atoms in the matrix, the larger the $I_D/I_G$ value. We find that the introduction of Cu nanoparticles did not promote graphitization of the DLC surface compared with the pure PAO lubrication.

Therefore, The Raman results reveal that the sp$^2$ carbon structure of the DLC coating surfaces become more ordered during the friction process.

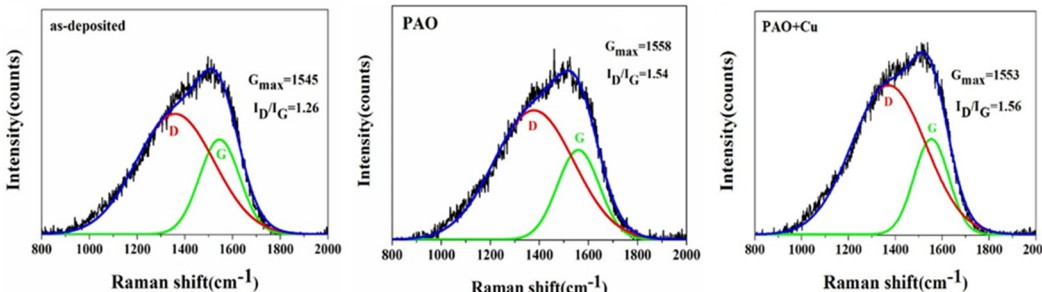

**Figure 11.** Raman spectra and fitting results of steel/DLC friction pair lubrication with PAO6 and PAO6 + 0.1 wt.% Cu.

Figure 12 shows the distribution of EDS elements on the wear track surface. The presence of elemental Cu on the wear surface under PAO6 + 0.1% wt.% Cu lubrication demonstrates the nano-filling mechanism and the "micro-bearing" mechanism. Results above indicate that the metal Cu nanoparticles form a "micro-bearing" between the friction pairs and the nano-filling mechanism mixes to change sliding friction into rolling friction during the friction process, so that the wear resistance was improved by avoiding friction pairs direct contact with each other.

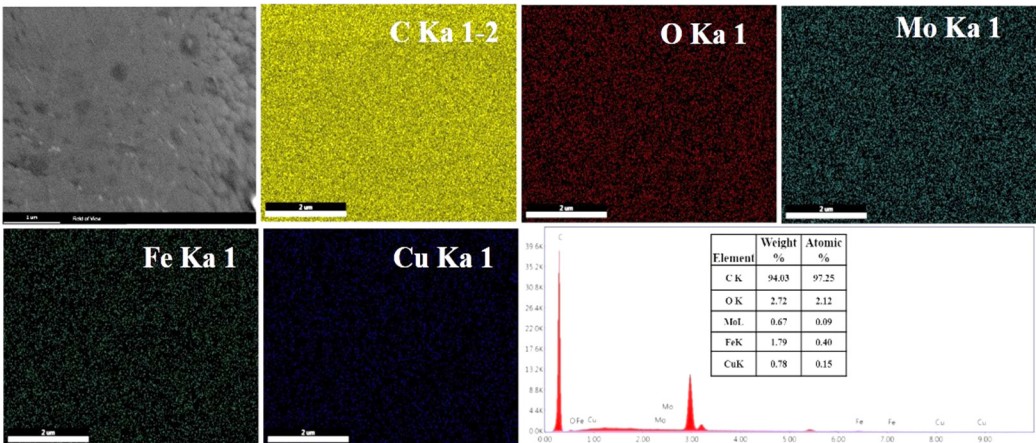

**Figure 12.** EDS results of wear track of steel/DLC friction pair under PAO6 + 0.1 wt.% Cu lubrication.

## 4. Conclusions

In this paper, metal Cu nanoparticles were prepared by an innovative technology called mass-produced nanoparticles (MPNP) and the tribological behaviors and lubrication mechanism of Cu nanoparticles as lubricant additives in DLC coatings were investigated. The main conclusions were drawn:

- The Cu nanoparticles fabricated by the macro-preparation technique have a core-shell structure with good spherical shape, high sphericity, mono-dispersion between particles and no agglomeration.
- Cu nanoparticles are used as anti-wear additives in lubricants to slightly decrease the friction coefficient and significantly reduce the wear rate of steel/DLC friction pairs, effectively providing wear protection for the surface of the friction pairs. Moreover, the optimized content of metallic Cu nanoparticles in the PAO6 base oil with 0.1 wt.% showed the lowest friction coefficient and wear rate, so the best anti-wear and friction reduction performance was achieved, while the friction coefficient and wear rate increase slightly with the addition above 0.1 wt.%.

- Nanoparticles do not easily reside on the surface and shear to form a lubricant film due to the low adhesion properties of DLC coatings. The tribological mechanism of the nanoparticles is mainly dependent on their rolling at the contact interface of the DLC coating. In addition, Cu nanoparticles reduce friction in steel/DLC friction pairs through nano-bearing action, and reduce stress concentration and wear in friction pairs through the nano-filling effect. The above results provide a new approach to the development of environmentally friendly additives for carbon-based coatings in the solid-liquid composite lubrication systems.

**Author Contributions:** Conceptualization, N.L.; Data curation, N.L.; Formal analysis, N.L.; Investigation, M.W.; Methodology, N.L.; Supervision, M.W.; Validation, M.W.; Writing—original draft, N.L.; Methodology, Z.W. All authors have read and agreed to the published version of the manuscript.

**Funding:** This research received no external funding.

**Data Availability Statement:** Data presented in this article are available at request from the corresponding author.

**Acknowledgments:** The authors would like to thank Zhiguo Wu of Institute of Nanomaterials Application Technology, Gansu Academy of Science for providing nano copper samples and experimental tests.

**Conflicts of Interest:** The authors declare no conflict of interest.

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
