# Peer review of "Mass-Produced Cu Nanoparticles as Lubricant Additives to Enhance the Tribological Properties of DLC Coatings"

_metals, doi:10.3390/met12081350_

Round 1

Reviewer 1 Report

All of the acronyms used in the text, e.g. Cu, DLC, GMO, EDS, Cr, PAO,  have to be explained.

The conditions of SEM analysis have to be indicated.

The is no information concerning the materials used in the procedures.

What is 304L?

Provide the scheme of the obtained core-shell structures.

The Authors indicated that “First of all, the substrate was ultrasonically cleaned with acetone and alcohol for 15 minutes and dried with dry nitrogen, then put them on the sample holders of the vacuum chamber.” (Lines 108-110) Please explain which of the substrates did the Authors have in mind.

The deposition of DLC coatings is unclear, please present it in the form of a scheme.

Line 161: “As shown in Figure 2, the DLC coating thickness of approximately 161 2600 nm is observed.”  First of all the designation should read Figure 4, not Figure 2. Secondly, please mark the thickness of the DLS layer on the SEM image.

Figure 4. What is a ”C layer”?

Figure 5. What is a “COF”?

There is no reason to divide the discussion into two parts: “3. Results and discussion” and “4. Discussion”. They have to be merged.

Lines 248-250: “Figure 9 (b-g) displays a significant reduction in the roughness of wear surfaces, and the number of furrows decreased and a smoother surface with the addition of different concentrations of Cu nano-additives.” Please explain in comparison with what the reduction of roughness is observed.

Please provide the values of Rq, Ra, and Rmax for all the studied surfaces and statistical analysis.

Author Response

The specific answers is shown in the file of “Response to Reviewer 1”.

Reviewer 2 Report

The paper contains an interesting and original study aimed at influence of Cu nanoparticles on the tribological properties of DLC coatings. The article is well organized and clearly written. The used methods were appropriately chosen for this kind of materials. The results are well described. A thorough discussion was provided. The paper may be published in “Metals” journal after moderate English quality check by a native speaker as it contains some grammar errors. One of the examples is: “Owing to the ion beam can deliver high energy continuously (…)“

Author Response

Thanks for your careful review and high affirmation. We carefully examined the language of the paper and asked for help from an engineer who studied abroad in the United States. The corresponding amendments are marked in red in the revised manuscript.

Reviewer 3 Report

The paper ”Mass-produced Cu Nanoparticles as Lubricant Additives to Enhance the Tribological Properties of DLC Coatings” is suitable for publication in Metals Journal with some minor corrections. First, the authors should improve the introduction with some coating elements' influence over the mechanical properties and corrosion properties. Please specify the SEM parameters of characterization. Please add the ICDD codes to the XRD analysis. In rest is ok.

Author Response

The specific answer is show in the file of “Response to Reviewer 3”.

Reviewer 4 Report

I think that your work is a considerable contribution to a topic of great industrial importance and therefore I think merits publication. Please allow me some suggestions for improvement:

1. It is not specified how many tests were performed for each concentration.

2. Fig. 5.a is the evolution of friction coefficient of samples, and Fig. 5.b shows the average values of friction coefficient. Did you obtain the average values of COF from Fig. 5.a? As shown in Fig. 5.a, COF values are not constant and it changes. Therefore, the average value should be compared with stand deviation. In Fig. 5.b, the standard deviation is missing. As shown in Fig. 5.a, the COF changes in time, and the average values should be provided with deviation.

3. Figures 9 (a-g) are very small. Use the same scale. "A significant reduction in roughness of wear surfaces" - Is the roughness reduction specified only qualitatively, without any roughness parameter?

Author Response

The specific answers is show in the file of “Response to Reviewer 4”.

Round 2

Reviewer 1 Report

The section “Preparation and characterization of nano-Cu particles” is still unclear. There is no information concerning the producer and used amount or condition used in the procedures.

The scheme of the obtained core-shell structures (text and figure) should be presented in the manuscript, not only in response to the Reviewer.

The Same in the case of the comment presented below: The Authors indicated that “First of all, the substrate was ultrasonically cleaned with acetone and alcohol for 15 minutes and dried with dry nitrogen, then put them on the sample holders of the vacuum chamber.” (Lines 108-110) Please explain which of the substrates the Authors have in mind.

There is still a lack of statistical analysis concerning Rq, Ra, and Rmax

Author Response

See the attachment for specific point-to-point answers.

This manuscript is a resubmission of an earlier submission. The following is a list of the peer review reports and author responses from that submission.